# Solution-Processed Large-Area Organic/Inorganic Hybrid Antireflective Films for Perovskite Solar Cell

**DOI:** 10.3390/molecules28052145

**Published:** 2023-02-24

**Authors:** Mingfang Huo, Yun Hu, Qin Xue, Jingsong Huang, Guohua Xie

**Affiliations:** 1Department of Physical Science and Technology, Central China Normal University, Wuhan 430079, China; 2Oxford Suzhou Centre for Advanced Research (OSCAR), University of Oxford, Suzhou 215123, China; 3The Institute of Flexible Electronics (Future Technologies), Xiamen University, Xiamen 361005, China

**Keywords:** antireflection film, organic/inorganic hybrid material, adjustable refractive index

## Abstract

In recent years, organic/inorganic hybrid materials have attracted much attention in the field of multilayer antireflection films because of their excellent optical properties. In this paper, the organic/inorganic nanocomposite was prepared from polyvinyl alcohol (PVA) and titanium (IV) isopropoxide (TTIP). The hybrid material has a wide, tunable window of refractive index, i.e., 1.65–1.95, at a wavelength of 550 nm. The atomic force microscope (AFM) results of the hybrid films show the lowest root-mean-square surface roughness of 2.7 Å and a low haze of 0.23%, indicating that the films have good potential for optical applications. The double-sided antireflection films (10 × 10 cm^2^) with one side of hybrid nanocomposite/cellulose acetate and the other side of hybrid nanocomposite /polymethyl methacrylate (PMMA) achieved high transmittances of 98% and 99.3%, respectively. After 240 days of aging testing, the hybrid solution and the antireflective film remained stable with almost no attenuation. Furthermore, the application of the antireflection films in perovskite solar cell modules increased the power conversion efficiency from 16.57% to 17.25%.

## 1. Introduction

Organic/inorganic hybrid materials could be solution-processable and have superior optical properties, mechanical strength, and thermal stability [1,2,3]. It is feasible to modulate the antireflective properties of the multilayer hybrid films. The working wavelengths of the antireflection films range from the deep ultraviolet to the near-infrared region, covering the entire visible region. Therefore, many applications could be implemented, including optics, communication, and photovoltaics [4,5]. Due to the complexity of materials and processes, large-area antireflection films are not widely studied [6].

For preparing the antireflective films, the main characteristics of the prepared material to be considered are transparency, refractive index, ease of processing, cost, mechanical properties, solubility, and biosecurity. The single-layer, large-area antireflective films mostly require surface structure modification or a complicated nanostructured surface from the templates [7,8]. The multilayer antireflective films are mostly prepared using magnetron sputtering, vacuum vapor deposition, and pulsed laser deposition, especially for the inorganic materials such as TiO_2_, Al_2_O_3_, HfO_2_, MgF_2_, and SiO_2_ [9,10,11]. These approaches usually have the disadvantage of being complicated and requiring high temperature (≥400°C) treatment. Alternatively, organic/inorganic hybrid materials fabricated by solution processes are applicable as antireflective films because of their lower cost and excellent optical properties [3,12]. The organic/inorganic hybrid system of titanium oxide hydrate/polyvinyl alcohol (PVA) was first proposed by Manuela Russo [13], which was used to prepare distributed Bragg reflectors (DBR) due to its excellent optical properties [14,15].

In the organic/inorganic hybrid system, the preparation process was complicated when TiCl_4_ was used as a precursor. Moreover, the stability of the hybridization solution is poor. For instance, only after one day would the prepared films release acidic gas and change dramatically [13,14,16]. When TiBr_4_ was used as a precursor, although the chemical cleaning method was used instead of annealing, the refractive index of the hybrid material had a small range of tunability [17]. Titanium (IV) isopropoxide (TTIP) was frequently used as a TiO_2_ precursor to prepare antireflective films [11], which usually required high processing temperatures. In some cases, TTIP was used as a precursor to prepare the hybrid materials, which can be prepared quickly and mildly at room temperature. In addition, the prepared films did not produce acidic gases [18]. In order to obtain a clarified and stable solution, it typically required two chelating agents, i.e., acetylacetone (HAcAc) and acetic acid (AcOH), to improve stability when preparing the TiO_2_ films by TTIP as a precursor. This is because the presence of both chelating agents produces a certain amount of polynuclear titanium [19]. However, this would result in a decrease in mononuclear titanium and, as a result, a deterioration in stability [13,14,17].

In this contribution, we prepared the organic/inorganic hybrid material PVA-TTIP (PT for short) using vinyl alcohol (PVA) and titanium (IV) isopropoxide (TTIP) as the raw materials and acetylacetone and acetic acid as the stabilizers. To reduce the generation of the multinucleated titanium, the hybrid materials were prepared, and acetic acid was added to improve the stability. The refractive index of the hybrid film can be easily modulated in the range of 1.65–1.95 at a wavelength of 550 nm, and the film haze is only 0.23%. The PT/Cellulose Acetate (CA) and PT/Polymethyl Methacrylate (PMMA) double-sided antireflection films were designed and prepared on a large area (10 × 10 cm^2^) glass substrate with a high transmittance of 98% and 99.3%, respectively. After aging for 240 days, the hybrid material solution remained stable, and the antireflection films prepared with the hybrid solution still possessed excellent optical properties without attenuation. Furthermore, antireflective films were applied to photovoltaic devices to improve performance by reducing surface reflection [20]. Perovskite solar cells are attracting global attention due to their low cost, simple process, abundance of ingredients, and high power conversion efficiency [21]. For commercial applications, it is essential to maximize efficiency and stability [22]. Typically, the glass encapsulation and the reflective metal contact of perovskite solar cell led to the optical loss of sunlight incoupled into the active layer and thus deteriorate the overall photovoltaic performances. The hybrid antireflection films demonstrated in this investigation were used in the cover glass of a perovskite solar cell, which increased the power conversion efficiency from 16.57% to 17.25%.

## 2. Results and Discussion

### 2.1. Theoretical Simulations of the Antireflective Films

The optical simulations were performed before fabricating the film stacks to meet the desired transmittance and provide guidance for further optimization. The transmittance of the air-film interface could be obtained according to the simplified Fresnel equations [5].
(1)$$R={\left[\frac{n_{air}-n_{film}}{n_{air}+n_{film}}\right]}^{2}\;$$
(2)$$T=1-R=\frac{4n_{air}n_{film}}{{\left(n_{air}+n_{film}\right)}^{2}}a=1$$

$n_{air}$ represents the refractive index of air and $n_{film}$ represents the refractive index of a thin film. The transmittance of a multilayer film at positive incidence, without considering optical effects such as absorption and scattering, can be calculated using the characteristic matrix equation derived from the above equations [23]. Cellulose acetate (CA) and PMMA were commonly used materials for the preparation of the optical thin films, which were inexpensive and highly transparent [4,5,14]. When CA and PMMA were used as the low refractive index materials for the antireflective films, the optical properties of the bilayer structures of the hybrid material/CA stack and the hybrid material/PMMA stack were simulated by the commercial software Setfos (Fluxim AG, Swithzerland), respectively. We first testified about the designed film stack architectures shown in the insets of Appendix A, respectively (see Appendix A). The maximum transmittance of 94.7% and 95.8% were achieved at 550 nm, respectively, when the thicknesses of the hybrid films were 80 nm and 81 nm on the glass substrate with a refractive index of 1.5045, respectively (see Appendix A).

### 2.2. Chemical Properties of the Hybrid Material

The preparation of organic/inorganic hybrid materials requires two processes. The first process is the modification of TTIP to obtain an inactivated complex, and the second one is the cross-linking of PVA with the modified TTIP. The chemical bond and functional groups of the hybridized material PT (14.4%) were evaluated by a Fourier transform infrared spectrometer (FTIR), as shown in Figure 1a. The sharp peaks at 880, 1049, and 1381 cm^−1^ indicated free alcohol groups in the material. In contrast, the broad peaks below 900 cm^−1^ indicated the Ti-O-Ti network structure in the material [18,19]. Since TTIP was produced in the hydrolysis and condensation reactions with the different contents of alcohols, the peak of PT tended to be stronger below 900 cm^−1^. The peak at 1090 cm^−1^ accounts for the methyl group in PVA. Meanwhile, the methyl group of HAcAc was bound to Ti in TTIP-sol, and the hybrid material PT contained the methyl group of AcOH bound to Ti [19], resulting in a higher peak intensity. The peak at 1527 cm^−1^ implied the C=O bond of HAcAc bound to Ti in TTIP-sol [24,25]. Nevertheless, the structureless peak at 3365 cm^−1^ resulted from the vibration of Ti-OH bond and -OH [25]. The peak intensity of the hybrid material PT was stronger here after the reactions were completed. Overall, the FTIR spectra of the two chelators could accurately describe the structure of the modified TTIP [18]. Polynuclear titanium was usually produced during annealing and aging as the complexes left the mononuclear titanium structure while the polymer content was low. It is due to the covalent bonds formed between the polynuclear titanium and the polymer [13,14,17]. Based on the source structure of the organic/inorganic hybrid system of titanium oxide hydrate/PVA [26], the cross-linked structure of the hybridized materials is schematically shown in Figure 1b. It is found that the viscosity of PT solution remains almost unchanged after 240 days of aging (see Appendix A).

### 2.3. Optical Properties of the Hybrid Materials

According to the Bruggeman effective medium theory, in organic/inorganic hybrid materials, the higher the content of inorganic materials, the higher the refractive index. It is linearly related to the volume fraction of each component [27]. In order to obtain the optical properties that meet the requirements, finer tuning is required. The refractive index of the hybrid material film was simulated by the transmission matrix method, and the refractive index also affects the depth of transmission oscillation of the film [14]. The refractive index of the hybrid material increased with the increasing inorganic material content, i.e., from 1.82 to 1.94 at a wavelength of 550 nm (see Appendix A). The variations of the thickness and refractive index of the hybrid material PT (volume ratio of 14.4%) film with the concentration of PVA at the wavelength of 550 nm are shown in Figure 2a. It can be seen that the refractive index of the hybrid material decreased with the increase in PVA content. Although the low-refractive-index polymer material has less effect on the effective refractive index of the hybrid material, it changes the thickness of the film significantly. The refractive index and thickness of the hybrid material PT (14.4%) with PVA (2.5 mg/mL) were in line with the theoretical calculation shown in Appendix A. Therefore, the subsequent experiments were conducted at this ratio.

The refractive index and thickness of the films were measured and fitted by the F40-XER equipment. The variation of PT (14.4%) film thickness d, the optical path length nd, and the refractive index n with the annealing temperature are shown in Figure 2b. The refractive index variation of the hybrid material was mainly influenced by its thermal response properties of the hybrid material. As shown in Figure 2b, the film thickness decreased and the refractive index increased as the temperature increased, and the optical path length was not constant, which should be taken into account when designing the multilayer films. The hybrid material with different ratios had a large tunable range of the refractive index at the annealing temperature of 50–150°C, i.e., 1.65–1.95 (see Appendix A). The wavelength-dependent optical constants were measured with ellipsometry in order to accurately fit the optical properties of the antireflective films. For instance, the refractive indices of the hybrid material PT (14.4%) and the polymeric materials CA and PMMA were 1.85, 1.49, and 1.45, respectively, at the wavelength of 550 nm (see Appendix A).

Absorption is an important optical parameter of optical films. The absorption of the dielectric material affects the wavelength dispersion, and haze is used to determine the scattering effect caused by the interior and surface of the material. The hybrid PT film has a lower absorption than the TTIP-sol film in the range of 200–400 nm (Appendix A). The PT film’s haze was tested at only 0.23%, again indicating that the film has very low scattering. The atomic force microscope (AFM) measurement technique provides a quantitative comparison of the film surface roughness. The AFM plots of PT (14.4%) films at the areas of 2 μm × 2 μm and 10 μm. × 10 μm render the RMS roughness of the films, i.e., 2.7 and 6.4 Å, respectively (Appendix A). The roughness of the films at a smaller size was similar to that of bare glass [28]. The scanning electron microscope (SEM) image of the PT (14.4%) film presents a uniform surface morphology (see Appendix A). The very smooth films processed by spin-coating guarantee optical accuracy and reduced optical loss. It can also provide better optical interference in the multilayer film structure and improve the quality of antireflection.

### 2.4. Preparation of the Antireflective Films

The precise thickness of the film is the key to the prepared antireflective film based on the optical properties of the hybrid material. The CA/PT/CA structure yielded a maximum transmittance of 95.2% at a wavelength of 550 nm for the single-sided, triple-layer antireflective film (2 × 2 cm^2^) (Appendix A). The lower boiling point of acetone led to uneven thickness in the prepared films. However, the use of acetonitrile as the solvent resulted in some micro dot-like structures. DMAc as the solvent provided the smooth surfaces of the antireflective films (Appendix A). The PMMA/PT/PMMA structure had a maximum transmittance of 95.4% at 550 nm (Appendix A). The CA/PT/CA and PMMA/PT/PMMA structures had an average transmittance well above 90% in the wavelength range of 450 to 750 nm, respectively. 

The maximum transmittance of the PT/PMMA film (10 × 10 cm^2^) was up to 95.4%, which was slightly higher than that of the PT/CA film (see Appendix A). The transmission spectra of the prepared films and the fitted transmission spectra did not perfectly match each other in the low and high bands, which was very concerning due to the inevitable experimental errors. Moreover, it was partially caused by the local optimization of the refractive index at a single wavelength. Global optimization is time-consuming and challenging. The transmittance spectra of the PT/PMMA and the PT/CA films (2 × 2 cm^2^) prepared on quartz substrate matched well with the fitted spectra for this structure (see Appendix A). The haze of bi-layer and triple-layer antireflective films was close to 0. The transmittance spectra of large-area and double-sided PT/CA and PT/PMMA antireflection films are shown in Figure 3a. The double-sided film with PT/PMMA structure exhibited a rather higher peak transmittance of 99.3%, which was 0.6% lower than that of the simulated result and 1.3% higher than that of the double-sided film with PT/CA structure. Nevertheless, the double-sided film with the PT/CA film still achieved a reasonable transmittance of 98%.

The transmittance spectra of the film aged at different time intervals were measured (see Figure 3b). It can be observed that the transmittance of the film at the target wavelength was maintained at ~98%, which increased slightly with the aging time. This might be caused by the evolution of the refractive index of PT. The single-sided antireflective films with PT/CA film under the same preparation conditions were prepared with the hybrid material solutions aged at different time intervals, as shown in Figure 3c. The peak transmittance of the film was consistently maintained at 95.2%. The results of these two aging experiments show that the films and solutions prepared from the hybrid materials have very high stability for practical applications. The large-area uniformity test results are shown in Figure 3d. The thickness of the film was measured using a probe-type profilometer. The thickness variation of the film was kept within a range of 5 nm. The reflection images of the double-sided antireflective films prepared with PT/CA and PT/PMMA structures are shown in Figure 4, demonstrating the excellent performance of the antireflective films.

### 2.5. Application to Photovoltaic Modules

The photovoltaic performance of the perovskite solar cell was measured under the global AM 1.5 spectrum (100 mW/cm^2^) to attest to the antireflective effect of the films. The PT/CA antireflective film applied to the perovskite solar cell had a positive effect on device performance, as shown in Table 1. The current density-voltage (J-V) curves and EQE characteristics of the perovskite solar cell are shown in Figure 5. It can be observed that the current density of the perovskite solar cell was increased by 0.34 and 0.60 mA/cm^2^ after attaching the single- and double-sided antireflective films to the device, which led to the increase in PCE from 16.57% to 16.85% and 17.25%, respectively, i.e., an improvement factor of 1.7% and 4.1%, respectively. It was noticed that the performance of the perovskite solar cell with an antireflective film prepared on the cover glass decreased in the wavelength range of 300–400 nm due to self-absorption (see Appendix A). The antireflective films with PT/PMMA structures have the same trend on the perovskite solar cell (Appendix A).

## 3. Materials and Methods

### 3.1. Materials

Titanium (IV) isopropoxide (TTIP, 97%, ρ= 0.96 g/mL at 20°C (lit.)), polyvinyl alcohol (PVA, weight average molecular weight M_w_ ~ 30,000 g/mol), cellulose acetate (CA, number average molecular weight M_n_ ~30,000 g/mol), polymethyl methacrylate (PMMA, mass average molecular weight M_w_ ~35,000 g/mol), and N, N-Dimethylacetamide (DMAc, 99.8%, anhydrous) were all purchased from Sigma-Aldrich (Shanghai, China). Acetylacetone (HAcAc, 99%, ρ= 0.975 g/mL at 25°C (lit.)), Acetic acid (AcOH, 99%, ρ= 0.975 g/mL) were purchase from Adamas (Shanghai, China). Acetone (99.5%) and n-butyl acetate (BAC, 99.5%), were purchased from Greagent (Shanghai, China). N, N-Dimethylformamide (DMF, 99.5%, anhydrous) and acetonitrile (99%) were purchased from Sinopharm Chemical Reagent Co., Ltd (Shanghai, China). The precursors, e.g., CsI, FAI, PbBr_2_, MABr, and PbI_2_, used to construct the light-absorbing perovskite active layer, were purchased from Xi’an Polymer Light Technology Corporation (Xi’an, China). All the materials were used as received, without any purification.

### 3.2. Preparation of Thin Films and Solutions

The TTIP solution was slowly added to HAcAc to meet the molar ratio of TTIP (HAcAc = 1:6). The solution was stirred for 10 min at room temperature. The film prepared from the orange-yellow solution was recorded on TTIP-sol film. The DMF solution of PVA was completely dissolved after heating and stirring at 90°C for 6 h and was then filtered using a filter (PTFE, pore size 0.45 μm, Titan, Shanghai, China). 2 mL of the filtered PVA solution (2.5 mg/mL) and 8 mL of the modified TTIP-sol solution were mixed and stirred at room temperature for 3 h. To ensure the stability of the solution, 1.5 mL of acetic acid was added and stirred at room temperature for 6 h. The stirring after several hours resulted in the cross-linking of the hydroxyl groups of PVA with the modified titanium complex intermediate, which led to the hybridized structure. Finally, a yellow clarified solution was obtained with a pH value of 4.65, which is called the PT solution. Only the TTIP-sol ratio was modified in order to configure the other PT solution ratios.

The hybrid PT films were prepared by spin-coating, and the preparation process of large-area PT/CA structured double-sided antireflection films is shown in Figure 6. Cellulose acetate was dissolved in three solvents, i.e., acetone, acetonitrile, and DMAc, respectively. For PMMA, n-butyl acetate was used as the solvent. The polymeric material was stirred for 6 h, then filtered with PTFE (0.45 μm). The deposited films were generally spin-coated at a speed of 1500 rpm for 30 s, followed by stepwise annealing at 100 and 150°C for 15 min.

The thickness of the low-refractive-index material film was mainly controlled by the concentration and the speed of spin-coating. When DMAc was used as the solvent for CA, the solution viscosity was 6.63 mPa∙s, which was relatively high for the spin-coating method. Therefore, the solution was sonicated for 15 s to remove the tiny air bubbles in the solution before preparing the large-area film, and the multi-step spin coating method was implemented to deposit the film, which made the prepared film more uniform.

### 3.3. Morphology and Optical Characterization

Fourier transform infrared spectrometry was measured with the IRAffinity-1S (Shimadzu, Kyoto, Japan). The surface topography of the films was obtained using an atomic force microscope (AFM—Dimension Icon, Bruker, Berlin, Germany) and an Olympus microscope (BX63, Olympus, Allentown, PA, USA). SEM cross-sectional morphology was taken on a field emission scanning electron microscope (Nova NanoSEM450, Nebraska, NE, USA). The refractive indices of the films were measured and fitted by a spectroscopic ellipsometer (M2000D, J.A. Wollam, Lincoln, NE, USA) and a Filmetrics contactless film thickness gauge (F40-EXR, A KLA, San Diego, CA, USA), respectively. The transmission spectra of the films were measured by a UV spectrophotometer (UV-2600, Shimadzu, Koto, Japan). The haze of the film was measured by a haze meter (TH-100, CHN Sper, Taizhou, China).

### 3.4. Fabrication and Characterization of Perovskite Solar Cells

The perovskite solar cell was composed of indium tin oxide (ITO)/ poly[bis(4-phenyl)(2,4,6-trimethylphenyl)amine](PTAA)/perovskite active layer/[6,6]-phenyl-C61-butyric acid methyl ester (PCBM)/bathocuproine (BCP)/Al. PTAA was spin-coated onto the ITO substrate from a chlorobenzene solution (3 mg/mL). PbI_2_ (1.36 mmol), FAI (1.26 mmol), PbBr_2_ (0.14 mmol), and MABr (0.14 mmol) were dissolved in DMF and DMOS mixed solvent (4:1, v/v, 1 mL). After stirring the solution for 1 h, CsI in DMOS (40 μL, 2 mol/L) was added to the above precursor solution. The solution was stirred again until it was clear before use. The perovskite film was formed after spin-coating the solution onto PTAA and was simultaneously quenched by the anti-solvent chlorobenzene 8 s ahead of the end of the total 40 s spin-coating procedure. Later, a layer of PCBM (20 mg/mL in chlorobenzene) was spin-coated onto the active layer at 1000 rpm for 40 s, covered by BCP spin-coated from the isopropanol alcohol solution (0.5 mg/mL). Finally, the sample was loaded into a high vacuum chamber, and the Al cathode was thermally evaporated at a pressure of 1 × 10^−4^ Pa. The active area was defined by a shadow mask with a size of 2 mm × 2 mm.

The current-voltage characteristics of the perovskite solar cell were determined without encapsulation by a source unit meter (Keithley 2400, Tektronix, Beaverton, OR, USA) and a calibrated solar simulator. An interval of 10 mV and a delayed time of 10 ms were applied when collecting the data points. The external quantum efficiency of the perovskite solar cell was recorded by a Bentham quantum efficiency measurement equipment (PVE 300, Bentham Instruments Ltd, Reading, UK) calibrated with a commercially available silicon detector.

## 4. Conclusions

The organic/inorganic hybrid materials composed of polyvinyl alcohol and titanium (IV) isopropoxide as raw materials and acetylacetone and acetic acid as stabilizers were designed and prepared to construct the antireflective film for light incoupling in the perovskite solar cell. The refractive index of the hybrid material can be feasibly controlled in the range of 1.65–1.95 at a wavelength of 550 nm. We have prepared double-sided antireflective films using the organic/inorganic hybrid materials by spin-coating on a large-area glass substrate of 10 × 10 cm^2^. The PT/CA and PT/PMMA double-sided antireflective films have a high transmittance of 98.0% and 99.3% at the target wavelength of 550 nm, respectively. After aging for 240 days, the hybrid material solution and the antireflective film remained stable. Furthermore, the application of the fabricated antireflection films in the perovskite solar cell increased the power conversion efficiency by an improvement factor of 4.1%. The antireflective films prepared from the hybrid material PT have excellent optical properties and good stability. Considering the tunable refractive index range of the hybrid material, further improvements are easily envisaged by the deliberate optical design. Due to the intrinsic high absorbance of the organic/inorganic hybrid films stack in the UV region, it is expected that such a design should play a major role in preventing the UV-induced degradation, which would be systematically investigated in the future.

## Figures and Tables

**Figure 1 molecules-28-02145-f001:**
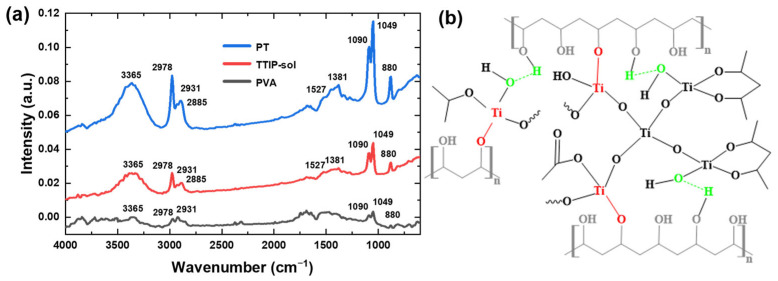
(**a**) FT-IR spectra of PVA, TTIP-sol, and PT. (**b**) Schematic structure of PT. Both mononuclear titanium (left) and Ti-O-Ti meshes (right) are represented, with hydrogen bonds (green) and covalent bonds (red) indicating a good, cross-linked structure.

**Figure 2 molecules-28-02145-f002:**
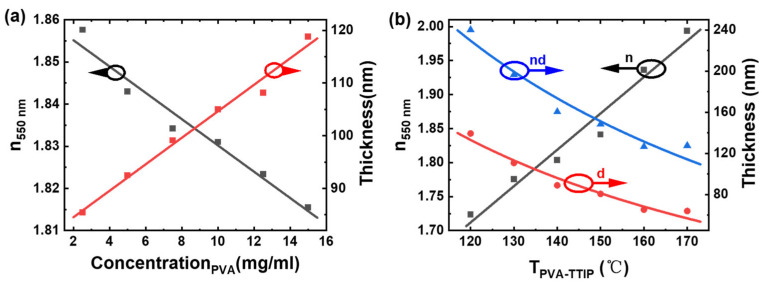
(**a**) The thickness (red curve) and refractive index (black curve) of the hybrid material PT (14.4%) films with PVA content at the wavelength of 550 nm. (**b**) The effective refractive index (n), the optical path length (nd), and the thickness (d) of the hybrid film PT (14.4%) spin-coated on glass substrates after annealing at different temperatures.

**Figure 3 molecules-28-02145-f003:**
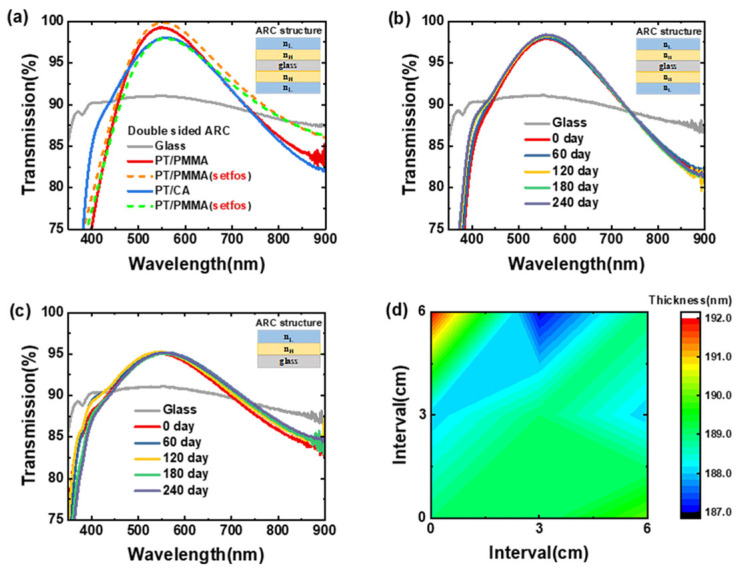
(**a**) Transmittance spectra of the double-sided antireflective films with large-area PT/CA and PT/PMMA structures. (**b**) Transmittance spectra of the large-area and double-sided antireflective films of PT/CA aged at different time intervals. (**c**) Transmittance spectra of the bilayer and single-sided films fabricated from the aged hybrid material solution at the same preparation conditions, taking the PT/CA structure as an example. (**d**) Thickness distribution contour plot of the single-sided and large-area antireflective films with PT/CA structure. n_H_: high-refractive-index hybrid PT film; and n_L_: low-refractive-index materials CA and PMMA at each side of the glass substrate.

**Figure 4 molecules-28-02145-f004:**
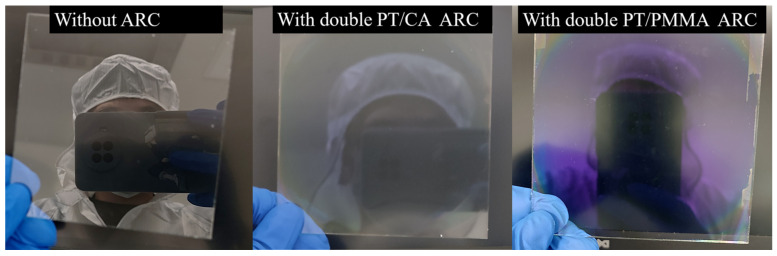
From left to right, the reflection images of the large-area (10 × 10 cm^2^) bare glass, a large-area and double-sided PT/CA film stack, and a large-area and double-sided PT/PMMA film stack.

**Figure 5 molecules-28-02145-f005:**
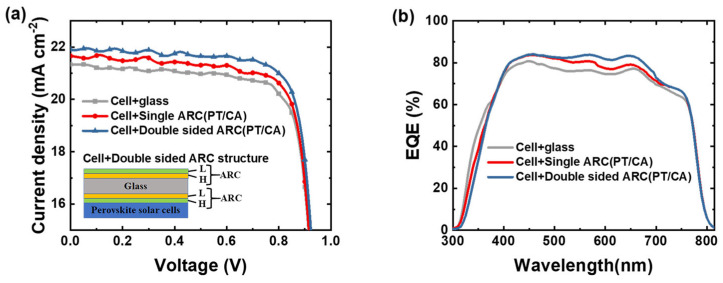
(**a**) J-V curves and (**b**) EQE plots. H: high-refractive-index hybrid material PT; and L: low-refractive-index material CA. The perovskite solar cell with the PT/CA structured single-sided and double-sided antireflective films was prepared and attached as the cover glass, respectively.

**Figure 6 molecules-28-02145-f006:**
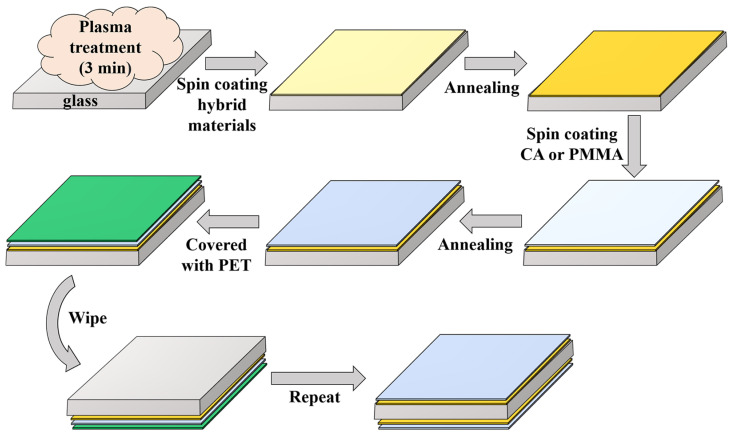
Fabrication flow of large-area double-sided antireflection film.

**Table 1 molecules-28-02145-t001:** Photovoltaic performance of perovskite solar cells before and after using the PT/CA antireflective films as the cover glass (perovskite composition: (CsPbI_3_)_0.05_[(FAPbI_3_)_0.9_(MAPbBr_3_)_0.1_]_0.95_).

Perovskite Solar Cell	J_SC_ (mA/cm^2^)	J_SC_ (EQE) (mA/cm^2^) ^a^	FF (%)	V_OC_ (V)	PCE (%)
With bare glass	21.32	18.91	79.91	0.97	16.57
With a single ARC (PT/CA)	21.66	19.68	79.99	0.97	16.85
With double-sided ARC (PT/CA)	21.92	20.02	80.35	0.98	17.25

^a^ Current density calculated from the EQE. The device structure of the inverted perovskite solar cell is ITO/PTAA/Perovskite/PCBM/BCP/Ag. The area of the active layer was 0.04 cm^2^.

## Data Availability

The data presented in this investigation is available from the corresponding authors.

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
