# Peer review of "Solution-Processed Large-Area Organic/Inorganic Hybrid Antireflective Films for Perovskite Solar Cell"

_molecules, 2023, doi:10.3390/molecules28052145_

Round 1

Reviewer 1 Report

The manuscript reports on the design of polymer layers for antireflection coatings of perovskite solar cells. The manuscript is in general well written and interesting results are presented. There are some minor grammatical issues that require consideration as highlighted in the attached pdf file. Schematics of the main layer designs are required in the main text to clearly illustrate the investigated structures. The major issues include a limited scope regarding the application of the coating in perovskite solar cells. In particular, prior literature on polymer layers for perovskite cells was not introduced and the significance of the presented stack is not clear. In fact, antireflection coatings are required in all types of cells, therefore, a more solid foundation for the proposed application is required. Moreover, the stability of the double-sided film structure is expected to be low due to exposure to the ambient and a discussion of this aspect is required. Detailed comments can be found in the attached pdf file.

Author Response

We gratefully thank the reviewer for the insightful suggestions and the critical comments, which dramatically improve the quality of our manuscript. Please find attached the point-to-point responses.

Reviewer 2 Report

In this manuscript, Huo et al. reported the fabrication of polyvinyl alcohol (PVA) and titanium isopropoxide (TTIP) organic/inorganic hybrid composites for antireflection films and their application in perovskite solar cells. Most parts of the paper focus on the properties of antireflection films, while their application and characterization in perovskite solar cells are very simple and rudimentary. Furthermore, despite the proposed hybrid antireflection film looks useful for perovskite photovoltaics, how they perform compared to other antireflection techniques is unclear. I would suggest a major revision of this manuscript before its publication. 

1.      The composition of perovskite is not provided. It seems MA, FA, and Cs were used with mixed halides (Br/I) but their atomic ratio in form of thin film was not confirmed. Also, it is not clear why the authors choose mixed-halide perovskite instead of typical mono-halide perovskites.

2.      Perovskites are ionic semiconductors, which may react with the surrounding environment (moistures, oxygen, etc.) or contact materials and deteriorate their long-term stability. The authors should test the compatibility between the hybrid nanocomposites and perovskites to confirm their potential for long-term usage.

3.      The EQE spectra of perovskite solar cells can be further analyzed to calculate the integrated current density and compare with experimental J–V curves (see ACS Energy Lett. 2022, 7, 12, 4150–4160 as an example).

4.      The authors are suggested to compare the performance of PT/CA with other antireflection films in perovskite solar cells to verify its practicality.

Author Response

The reviewer isgratefully acknowledged for the insightful suggestion and comments. Please find attahced our point-to-point responses to the comments.

Reviewer 3 Report

The authors reported on “Solution-processed large-area organic/inorganic hybrid antireflective films for perovskite solar cell” is quite interesting and such article can be accepted for the publication.

The authors clearly explained that the hybrid materials antireflective films need mild processing techniques compare to the inorganic materials.

The hybrid materials have high optical transmittance and used in perovskite solar cells to increase the photovoltaic performance. After aging for 240 days, the hybrid material solution and the antireflective film remained stable which is quite advantage to improve the stability of the perovskite solar cells.

The authors have prepared two kinds of hybrid antireflective materials and both of them showed improved performance. But PT/CA showed higher performance compared to the other one.

This kind of antireflective systems could be really useful of commercialization of perovskite solar cells by increasing the performance along with the stability.

The paper can be accepted in the present form.

Author Response

The reviewer is gratefully appreciated for the positive comments. This work provides a simple and cost-effective method to improve the performance of solar cells.

Reviewer 4 Report

This work deals with Solution-processed large-area organic/inorganic hybrid antireflective films for perovskite solar cells. Its review is very interesting work in the related field for potential readers. However, there is not clear something for publication and I recommend the author should improve the manuscript.

1. In Figure 3, your samples have higher transmittance ~ 99.9%. than normal glass. In the case of glass, the transmittance has around 90% in the range of 400-900 nm but after 550 nm, the transmittance of prepared samples was decreased by under 83%. Could you explain the reason?

2. Authors mentioned” large-area” in your work. How about uniformity in the papered samples? (thickness and transmittance in middle, top, and down of films).

3. If possible, I recommend you would show XRD, TEM, or FE-SEM images of samples for potential readers.

4. In the photovoltaic performance of cells, all samples have about 16~17 % of PCE. There is no enhancement in each sample.  Is it the right ARC effect for improving cell performance?

Author Response

We gratefully thank the reviewer for the useful comments to make our manuscript more acceptable for publicaiton. Please find attached the point-to-point comments.

Round 2

Reviewer 1 Report

The authors partially addressed the concerns of the review. However, adding replies to the concerns in the main text of the manuscript is required. Especially the device schematics, that aid towards a better understanding by the reader, should be presented clearly in the main and not just the supplementary.

Author Response

We thank the reviewer for the suggestion. The stacked architectures of the designed antireflective films are shown in Figure 1, inserted on Page 3. The following correction has been inserted in the main text.

"All the stacked architectures of the designed antireflective films investigated in this paper are shown in Figure 1, and we first testified the designed film stack architectures displayed in Figure 1b.".

The details of the corrections could be found in the revised manuscript and the response attached to this comment.

Reviewer 2 Report

The revision looks ok. I don't have further questions.

Author Response

We thank the reviewer for the positive comments.